# Metabolomic and Transcriptomic Profiling Identified Significant Genes in Thymic Epithelial Tumor

**DOI:** 10.3390/metabo12060567

**Published:** 2022-06-20

**Authors:** Enyu Tang, Yang Zhou, Siyang Liu, Zhiming Zhang, Rixin Zhang, Dejing Huang, Tong Gao, Tianze Zhang, Guangquan Xu

**Affiliations:** 1Department of Thoracic Surgery, Second Affiliated Hospital of Harbin Medical University, Harbin 150086, China; tey341204@163.com (E.T.); liusiyang202102@163.com (S.L.); 18846130858@163.com (Z.Z.); zrx15636106642@163.com (R.Z.); m17797735935@163.com (D.H.); gt18846837218@163.com (T.G.); tianzezhang1986@126.com (T.Z.); 2Department of Cardiac Surgery, Second Affiliated Hospital of Harbin Medical University, Harbin 150086, China; sheep930505@163.com

**Keywords:** thymic epithelial tumors, metabolomics, transcriptomic, metabolic score, prognosis, cancer metabolism

## Abstract

Thymomas and thymic carcinomas are malignant thymic epithelial tumors (TETs) with poor outcomes if non-resectable. However, the tumorigenesis, especially the metabolic mechanisms involved, is poorly studied. Untargeted metabolomics analysis was utilized to screen for differential metabolic profiles between thymic cancerous tissues and adjunct noncancerous tissues. Combined with transcriptomic data, we comprehensively evaluated the metabolic patterns of TETs. Metabolic scores were constructed to quantify the metabolic patterns of individual tumors. Subsequent investigation of distinct clinical outcomes and the immune landscape associated with the metabolic scores was conducted. Two distinct metabolic patterns and differential metabolic scores were identified between TETs, which were enriched in a variety of biological pathways and correlated with clinical outcomes. In particular, a high metabolic score was highly associated with poorer survival outcomes and immunosuppressive status. More importantly, the expression of two prognostic genes (ASNS and BLVRA) identified from differential metabolism-related genes was significantly associated with patient survival and may play a key role in the tumorigenesis of TETs. Our findings suggest that differential metabolic patterns in TETs are relevant to tumorigenesis and clinical outcome. Specific transcriptomic alterations in differential metabolism-related genes may serve as predictive biomarkers of survival outcomes and potential targets for the treatment of patients with TETs.

## 1. Introduction

Thymic epithelial tumors (TETs) are rare tumors of thymic epithelial origin located in the anterior mediastinum. World Health Organization (WHO) histological typing suggested that TETs can be categorized as thymomas (types A, AB, B1, B2, and B3), thymic carcinomas (TCs), and thymic neuroendocrine tumors [1]. Most patients are diagnosed after finding an anterior mediastinal mass by computed tomography (CT), and relatively few have clinical symptoms such as chest pain, cough, or manifestations of distant metastases; therefore, TETs need to be differentiated from thymic cysts, teratomas, and lymphomas due to imaging similarities. The thymus is the site of T-cell maturation and plays a central role in adaptive immunity, leading thymomas and thymic carcinomas to show variable degrees of thymus-like features; thus, TETs are commonly associated with paraneoplastic autoimmune diseases, including myasthenia gravis, pure red cell anaplasia, parathyroid adenoma, and hypogammaglobulinemia [2]. Surgery is the mainstay of treatment in the management of thymic tumors in terms of current treatment strategies, supplemented by radiotherapy and chemotherapy [3]. Most patients with TETs have a satisfactory prognosis after R0 resection and postoperative adjuvant therapy, which, however, is unattainable for patients who are unsuitable for R0 resection or have metastases [4,5].

There are still many pressing issues to be addressed in the paradigm of diagnosis and treatment for patients with TETs. (1) Pathogenesis remains unclear at the molecular level. Uncontrolled proliferation is a common feature of tumor cells, leading to altered metabolism because of the need for additional energy and biosynthetic precursors. However, the metabolic perturbations caused by the tumorigenesis of TETs have not been systematically described. (2) There is a lack of reliable diagnostic markers. Most TETs are diagnosed by CT alone and then operated on in the absence of pathological findings, resulting in some patients who lack the need for surgery, such as those with thymic cysts, being treated surgically for TETs. (3) Some patients suffer from concomitant symptoms. (4) In addition, patients who are unsuitable for surgery still demand effective alternative treatment options in the absence of sensitivity to radiotherapy and chemotherapy [6]. In summary, we need to explore the tumorigenesis of TETs and identify advanced treatment modalities.

Metabolites, being the cornerstone of cellular function, can be involved in enzyme-catalyzed chemical reactions and are essential for cellular function. The disruption of upstream biological information leads to timely and corresponding changes in metabolomics, which contains abundant information and is considered one of the most predictive phenotypes [7]. Untargeted metabolomics can help us measure the broadest range of metabolites in extracted samples without a priori knowledge of the metabolome and has been widely used for biomarker discovery in recent years. [8] A metabolomics strategy based on liquid chromatography–mass spectrometry (LC-MS) was used in this study to perform metabolic profiling on thymic cancerous tissues (TCTs) and adjunct noncancerous tissues (ANTs) of 32 patients with TETs. The comparison of metabolic features between tumor tissues and adjacent tissues was conducted first; subsequently, differential metabolism-related genes based on LC-MS results were filtered out, and the expression profiles of TETs were acquired from the cancer genome atlas (TCGA). The WHO histological type, which classifies TETs as clinically indolent (types A, AB, and B1; A/AB/B1) and aggressive TETs (types B1, B2, and TCs; B1/B2/C), has repeatedly been shown to be significantly associated with patient prognosis [9,10]; hence, we confirmed two subgroups (cluster 1 and cluster 2) using consensus clustering for differential metabolism-related genes (DMRGs) that stratified the WHO typing of TETs as well as survival status. Next, we developed a method to quantify metabolic patterns (a metabolic score) of individual tumors. The divided groups obtained by high and low metabolic scores had significant differences in survival and the immune environment. Importantly, we further investigated the involvement of two metabolism-related genes (MRGs), ASNS and BLVRA, in TETs. These findings may contribute to a better understanding of the unique pathogenesis of TETs and allow for maximum efficacy of genetic, cellular, and immune therapies.

## 2. Results

### 2.1. Differential Metabolism between TCTs and ANTs

A total of 1983 metabolic features were filtered out from our samples after data processing, of which 1237 were matched in the Human Metabolome Database (HMDB). Based on the profile of the identified metabolites, we transformed all samples into high-dimensional data points. Principal component analysis (PCA) was first performed to examine whether the data points were essentially distributed into two groups, and its score plots were represented by two principal components (Figure 1A). The results of the PCA showed that the data points were significantly clustered into two distinct groups based on TCTs and ANTs. In addition, the response intensities and retention times of the peaks for each quality control (QC) sample largely overlapped, indicating that there was negligible variation caused by instrument error throughout the experiment and that the stability and reproducibility of the investigation, as well as the reliability of the data quality, were sufficient for further analysis in this experiment. Partial least squares discriminant analysis (PLS-DA) was conducted, and clear differential clustering between TCTs and ANTs was consistently observed (Figure 1B). In addition, permutation tests (Figure 1C) also verified that the model was not overfitted (P(*R*^2^) < 0.001; P(*Q*^2^) < 0.001).

We first used univariate statistical analyses, including fold change (FC) analysis and Students’ *t*-test, to filter out the differentially expressed metabolic features between tumor and paraneoplastic tissues, which are presented in volcano plots with FC > 1.5 and *p*-value < 0.05 (Figure 1D). In the volcano plot, up- and downregulated metabolic features are marked in red and blue, respectively. We then performed pathway enrichment analysis on differential metabolic signatures and showed that the development of TETs may be related to the pathways of glycerophospholipid metabolism, arginine biosynthesis, and porphyrin and chlorophyll metabolism (Figure 1E, Appendix A). Furthermore, thymomas and thymic carcinomas were analyzed separately for metabolic differences from the corresponding normal tissues (Appendix A). As predicted, the metabolic pathways enriched in thymoma were almost similar to the former due to the larger proportion in this study, while the pathways significantly enriched in thymic carcinoma included only glycerophospholipid metabolism. In summary, we screened 32 pairs of tumors and paraneoplastic tissues for differential metabolic profiles, and the involved pathways may be associated with tumorigenesis.

### 2.2. Identification of Differential Metabolism-Related Genes in TETs with Prognostic Relevance

To identify differential metabolism-related genes (DMRGs) that are significantly associated with the prognosis of TETs, we first screened out genes relevant to metabolic differences between tumor and paraneoplastic tissues from a metabolite–protein interaction network (MPI) [11]. This network is composed of 1870 metabolites and 4132 proteins (represented by the encoding genes) from data sources, including the Kyoto Encyclopedia of Genes and Genomes (KEGG) [12], Reactome [13], Human-GEM [14], and BRENDA [15].

Subsequently, 931 DMRGs were selected for further analysis (Figure 2A and Appendix A). Samples from 117 patients with TETs in TCGA-THYM were then included in the weighted gene co-expression network analysis (WGCNA). In this study, a power of β = 4 (scale-free R2 = 0.87) was selected as the soft threshold (Appendix A). Then, the hierarchical clustering tree for 931 DMRGs was determined by conducting hierarchical clustering for dissTOM (Figure 2B), and we identified the gene units that were most associated with clinical features. In total, seven gene units were identified (Figure 2C). The brown module was found to have the strongest association with the WHO histological type and patient survival; thus, genes in this module were used for further analysis of the histological type and prognosis-related MRGs (TP-MRGs) (Appendix A).

### 2.3. Prognosis-Associated Metabolic Features Mediated by TP-MRGs

With the intention of further discovering the clinical relevance of the 137 TP-MRGs in the brown module, we first performed a gene ontology biological process and pathway enrichment analysis via Metascape (http://metascape.org/, accessed on 3 March 2022), an efficient and effective tool for comprehensive analysis by experimental biologists [16]. Not surprisingly, we found that most TP-MRGs were enriched in metabolic processes, including the monocarboxylic acid metabolic process, lipid catabolism, long-chain fatty acid metabolic process, amino acid metabolism, purine metabolism, and porphyrin metabolism (Figure 3A). These findings are consistent with the differential metabolism-related pathways described above. We then clustered the TET samples obtained from TCGA-THYM into subgroups based on the gene expression of TP-MRGs using the “ConsensusClusterPlus” R package. Based on the proportion of expression similarity and ambiguous clustering measures for the 137 TP-MRGs, k = 2 was determined to have the optimal clustering stability from k = 2 to 6 (Figure 3B; Appendix A). Accordingly, the 117 tumor samples were clustered into two subgroups (cluster 1: *n* = 43; cluster 2: *n* = 74). We analyzed the gene expression patterns between the two subtypes using principal component analysis (PCA) (Figure 3C), and we found that the gene expression profiles were well differentiated between the two subtypes. In addition, clinicopathological features were compared between the two subtypes (Figure 3D). Only TP-MRGs that were differentially expressed between the two subtypes (74/137) are shown in the heatmap. Cluster 2 tended to correlate with the aggressive histological type (B1/B2/C) (*p* < 0.05). These findings reveal that the clustered metabolic subtypes defined by TP-MRGs are closely associated with heterogeneity in patients with TETs.

### 2.4. Metabolic Score Construction and Its Clinical Relevance

Although we identified the role of different metabolic patterns in clinical prognosis based on the expression of TP-MRGs, these analyses are population-based and do not accurately predict the individualized metabolic patterns of tumors. We therefore used the LASSO Cox model to automatically screen for metabolic signature genes (MSGs) (Appendix A) and applied these genes to construct a scoring method (metabolic score) to quantify the metabolic patterns of individual TETs. Correlation analysis showed that the MSGs were all independent, and no collinear expression existed (Appendix A). An alluvial diagram was applied to visualize changes in the attributes of individual patients (Figure 4A). As shown in the alluvial diagram, most of the data streams in the B2/B3/C group originate from cluster 2, while the data streams in the high-score group mainly originate from the B2/B3/C group. Additionally, it was observed that patients who died in each of the three groups were clustered more into cluster 2, B2/B3/C, and high-score groups. These findings suggest that the above three factors may be associated with a poorer prognosis in patients with TETs. We next compared the differences in metabolic scores between the two subgroups by cluster and WHO histological type. Similar to the evaluation obtained from the alluvial diagram, cluster 2 and B2/B3/C were associated with higher metabolic scores in each subgroup, consistent with previous results in which the metabolic cluster was significantly associated with the prognosis-related WHO histological type (Figure 4B,C). As expected, patients with high metabolic scores were significantly associated with a worse prognosis in Kaplan–Meier analysis after samples lacking survival information were removed (*p* = 0.015, Figure 4D). Furthermore, multivariate Cox regression analysis indicated that the metabolic score was an independent predictor of overall survival in patients with TETs (Appendix A). We also explored the biobehavioral differences between the high- and low-metabolic-score groups by gene set variation analysis (GSVA) (Appendix A). As illustrated in the rainclouds, higher metabolic scores were significantly associated with the activation of metabolic pathways such as creatine, phenylalanine, and tyrosine metabolism. In conclusion, the metabolic score allows for a better assessment of the metabolic patterns of individual tumors.

### 2.5. Metabolic Patterns Characterized by Distinct Immune Landscapes

To further explore the correlation between the metabolic score and the biological behavior of this tumor, we compared the immunological profiles of the high- and low-scoring groups. As shown in the GSVA enrichment analysis (Figure 4E), a number of immune-related pathways were also present in the differential pathways between the high- and low-metabolic-score groups. In particular, the high-metabolic-score group was markedly enriched in the Fc gamma receptor pathway, the initial triggering of complement, antigen activation of the B-cell receptor leading to the generation of second messengers, and fcgr3a-mediated IL10 synthesis pathway. In addition, we applied the ESTIMATE algorithm to quantify the overall infiltration of the immune score, stromal score, and tumor purity between the two groups. In addition to the predictable higher tumor purity in the high-score group compared to that in the low-score group, we also found that the stromal score and the estimate score were significantly lower in the high-score group, while the immune score only showed a relatively, rather than significantly, lower tendency in the high-score group (Figure 5A–D). These results suggest that the proportion of immune infiltration was possibly lower in the high-score group than in the low-score group. Subsequently, we utilized CIBERSORT, a deconvolution algorithm for assessing the immune cell landscape in the tumor microenvironment using support vector regression, to assess immune infiltration. The results revealed that resting dendritic cells were significantly lower in the high-scoring group (Figure 5E). These results suggest that the high-scoring group had fewer non-tumor components (e.g., immune cells and stromal cells), probably because the downregulation of the immune infiltration proportion was more favorable for tumor development.

Based on the above evaluation of immune characteristics, the high-score group was classified as an immune-desert phenotype, characterized by a paucity of T cells in either the parenchyma or the stroma of the tumor. The low-score group, on the other hand, had the immune-excluded phenotype, featuring abundant immune cells retained in the stroma that surrounds nests of tumor cells [17]. Considering that CTLA-4 is a well-documented predictor of the response to anti-CTLA-4 treatment, we also compared CTLA-4 expression levels between the two groups and observed significant upregulation of CTLA-4 in the high-score group (Figure 5F). From the above results, we found that different metabolic scores are characterized by different immune infiltration, which may offer associations for tumor immunotherapy.

### 2.6. Expression, Prognosis, and GSEA of MSGs in TETs

Tumor-related metabolic alterations have functional consequences on the progression of the disease and survival outcomes [18]. We dichotomized the expression profiles based on the median expression of three previously selected MSGs and calculated the effect of high and low expression of each MSG on the overall survival of patients with TETs. We finally confirmed two genes that were significantly associated with survival in patients with TETs (Figure 6A,B). These two genes were ASNS, which encodes the glutamine-dependent asparagine synthetase, and BLVRA, belonging to the biliverdin reductase family, the members of which catalyze the conversion of biliverdin to bilirubin in the presence of NADPH or NADH.

Asparagine synthetase (ASNS) is an enzyme that catalyzes the conversion of aspartic acid to asparagine. This reaction requires glutamine as a nitrogen source and proceeds in an ATP-dependent manner [19]. Previous studies have demonstrated the involvement of ASNS in the tumorigenesis, treatment, and prognosis of a wide range of tumors, including human melanoma, breast cancer, and prostate cancer [20]. However, the expression levels and potential involvement of ASNS in TETs have not been investigated. High expression of ASNS, involved in amino acid metabolism, is associated with reduced survival in patients with TETs. Gene set enrichment analysis (GSEA) revealed that differential expression of ASNS in TETs was associated with pathways such as the adipocytokine signaling pathway, arachidonic acid metabolism, and arginine and proline metabolism (Figure 6C).

Overexpression of BLVRA has been reported in many diseases, including cancer. BLVRA expression has been shown to be elevated in many cancers, including liver, lung, breast, skin, and esophageal cancers [21]. Similar to ASNS, BLVRA upregulation has been associated with poorer prognosis in patients with TETs. Subsequent GSEA also suggested that differential BLVRA expression in patients with TETs was enriched in cancer pathways as well as pathways such as the phosphatidylinositol signaling system (Figure 6D).

To further confirm the extent to which each survival-related MSG of TETs affected final patient outcomes, we generated hazard ratios (HRs) for each gene as well as for varieties of clinical variables by univariate and multivariate analyses. As expected, the HRs for both ASNS and BLVRA suggested an increased risk of death (Table 1). Unfortunately, each of these variables was not an independent prognostic factor for patients with TETs. In summary, both ASNS and BLVRA are strongly associated with survival in patients with TETs.

## 3. Discussion

For nearly a century, tumors have been found to exhibit metabolic activities that distinguish them from well-differentiated non-proliferating tissues, which may contribute to their physiological survival and growth [22]. Cancer patients have been characterized as having several metabolic features, including deregulated uptake of glucose and amino acids, opportunistic patterns of nutrient acquisition, utilization of glycolytic/tricarboxylic acid cycle intermediates for biosynthesis, reduced nicotinamide adenine dinucleotide phosphate (NADPH) production, increased nitrogen demand, altered metabolite-driven gene regulation, and metabolic communication with the microenvironment [23]. With the help of these metabolic alterations, tumor cells can meet the needs of their own progression and adapt to changes in the tumor environment. Thus, the study of cancer metabolism can reveal fundamental aspects of malignancy and has the translational potential to improve the way that we diagnose, monitor, and treat cancer [24]. However, the metabolic profile of TETs has not been adequately studied and may offer new therapeutic options.

Due to the rarity of TETs, the number of cases included in metabolomic studies of patients with TETs is relatively limited. There was only a single study of metabolomic analysis in TETs by nuclear magnetic resonance spectroscopy, which included tissue samples from 15 TETs and 4 normal controls. It was suggested that pathways associated with cysteine, glutathione, lactate, and glutamine appear to be promising therapeutic targets based on metabolite differences between thymic epithelial tumor tissue and normal thymic tissue proline [25]. Inhibitors of glutaminolysis and the downstream tricarboxylic acid cycle (TCA) are also expected to be reasonable therapeutic strategies. In contrast, this study, employing liquid chromatography–mass spectrometry, expanded the sample size of tissue samples and screened for differential metabolic profiles between TCTs and ANTs, which were paired. The result revealed that the differential metabolism was enriched in glycerophospholipid metabolism and arginine biosynthesis. Glycerophospholipids are the most abundant phospholipids in eukaryotic cell membranes and play an important role in tumor development. Glycerophospholipid metabolism produces substrates that promote tumor cell proliferation and can regulate cell signaling pathways through downstream products [26]. Arginine, a versatile amino acid, is implicated in many metabolic pathways associated with tumorigenesis, including the synthesis of nitric oxide, creatine, and polyamines [27]. Levels of the rate-limiting enzyme for arginine biosynthesis, argininosuccinate synthase 1 (ASS1), are severely reduced or absent in some aggressive and chemoresistant cancers [28]. Thus, differential metabolism is involved in tumor development and is likely to contribute to the prognosis and diagnosis of tumors.

Metabolite–protein interaction networks bridge this study between metabolites and genes. To identify genes in DMRGs that could be prognostically relevant to TETs, we screened out 137 genes by WGCNA that were closely related to the WHO histological type and patient survival status. The tumor samples were grouped into two clusters by differential expression of the 137 genes, which were clearly differentiated in PCA. The heatmap (Figure 3D) also shows the differential expression profiles between the two clusters and correlates with WHO typing, suggesting that these 137 TP-MRGs are involved in the tumor development and clinical course of TETs. We used LASSO to further narrow down the 137 genes to 3 prognostic factors for TETs and constructed a metabolic score to quantify metabolic patterns. The metabolic score model was confirmed to be significantly different across clusters and WHO typing; Kaplan–Meier analysis suggested that high and low scores were significantly associated with survival status, while GSVA indicated differential metabolic patterns. The high- and low-metabolic-score subgroups had different immune phenotypes (immune-desert and immune-excluded phenotypes), which were associated with different anticancer immunity. The high-metabolic-score subgroup was classified as an immune-desert phenotype, associated with immune tolerance and minimal activation of cancer-specific T cells, leading to a worse prognosis. Unsurprisingly, this tumor phenotype rarely responds to anti-PD-L1/PD-1 therapy [29]. The immune-desert phenotype may reflect the absence of pre-existing antitumor immunity, suggesting that tumor-specific T-cell production is the rate-limiting step. Although the immune-excluded phenotype is characterized by the presence of abundant immune cells, the immune cells do not penetrate the parenchyma of these tumors but instead are retained in the stroma that surrounds nests of tumor cells. After treatment with anti-PD-L1/PD-1 drugs, stroma-associated T cells may show evidence of activation and proliferation but not infiltration, and clinical responses are uncommon. These results suggest that TETs may lack responsiveness to immunotherapy. However, elevated CTLA-4 levels in the high-metabolic-score group were significantly correlated, supporting the idea that immunotherapy may be of potential value for TETs. These findings may help to improve our understanding of the mechanisms underlying the tumorigenesis of different metabolic patterns in TETs and the immunotherapeutic potential of TETs.

Recent studies have shown that alterations in specific transcripts of metabolic genes can be used as predictive biomarkers of survival outcomes [30]. Similarly, our analysis found that the expression of two MSGs (ASNS and BLVRA) was predictive of survival for TETs.

ASNS proteins are widely expressed in different tissues and organs, but the basal expression of ASNS is relatively low in normal tissues, with the exception of the exocrine pancreas [31]. Because tumor cells have higher metabolic demands and they often grow in nutrient-deficient environments, ASNS transcription and translation are activated through a number of different mechanisms to protect cell survival. Inhibition of ASNS expression and subsequent asparagine depletion may reduce the proliferative capacity of tumor cells. Previous studies have reported that ASNS is involved in the tumorigenesis of various tumors. In breast cancer, knockdown of ASNS has been shown to inhibit cell growth by inducing cell cycle arrest [32]. In addition to this, ASNS has also been demonstrated to be involved in drug tolerance in prostate cancer and the prognosis of patients with hepatocellular carcinoma, respectively [33,34]. The present study is the first to identify the function of ASNS in TETs and may improve our understanding of tumorigenesis and clinical management in patients with TETs.

BLVRA has been found to be upregulated in both hepatocellular carcinoma and colorectal cancer and is expected to be a prognostic marker and potential therapeutic target [21,35]. Tumors not only require abundant fuel to maintain their rapid growth requirements but also demand an environment suitable for growth, including redox homeostasis. Biliverdin reductase (BVLR) is a well-characterized enzyme in the heme degradation pathway, which functions to convert biliverdin-IX-alpha into bilirubin-IX-alpha [36]. As the latter is a potent antioxidant, BLVRA, the major isoform of BLVR, is a pleiotropic enzyme and important in the maintenance of cellular redox homeostasis. In addition to catalyzing the aforementioned reactions, it can also modulate signal transduction, either directly through its serine/threonine/tyrosine kinase activities or indirectly by acting as a scaffold/bridge and intracellular transporter for kinases that mediate cell growth and proliferation. These diverse activities allow it to affect thousands of genes, including those involved in signaling pathways, apoptosis, and cyclins [35,36,37]. All of the above-mentioned factors suggest that high BLVRA expression has an important role in the tumorigenesis of TETs.

The main limitations of our study are the relatively small sample size, the monocentricity of the study, and the heterogeneity of the histotypes, all of which are associated with the extreme rarity of TETs, particularly TCs. In addition, although we matched a large number of metabolites through the HMDB, we were unfortunately not able to perform MS/MS to complete the metabolite annotation due to current laboratory constraints. However, the samples are stored intact in liquid nitrogen and will be reused for metabolite annotation in the future when improved conditions are available. In this study, while the combined utilization of innovative metabolite profiling for small population samples and the large, independent TCGA high-quality transcriptomic dataset brings about new insights into metabolic characteristics, it is necessary to validate our preliminary results in a prospective, larger collection of tumor material.

## 4. Materials and Methods

### 4.1. Metabolomics Sample Collection and Preparation

The protocol of this study was approved by the Ethics Committee of Harbin Medical University prior to the implementation of the study, including the collection of samples and personal clinical details. The TCTs and ANTs were obtained from 32 patients (18 males and 14 females, age range 32–80 years) (Table 2). All patients underwent surgery in the department of thoracic surgery at the second affiliated hospital of Harbin Medical University and were diagnosed with thymic epithelial tumors by pathologists.

The inclusion criteria for the patients were: (1) postoperative pathological confirmation of TETs; (2) no metabolic or immune diseases irrelevant to the tumor; (3) no previous history of malignancy; (4) normal indicators of the functional capacity of several critical organs and systems, such as liver and kidney. Exclusion criteria: (1) women who were menstruating, pregnant, or lactating; (2) lack of complete clinical information. Table 1 presents a summary of basic information on all patients, including sex, age, whether they had concomitant myasthenia gravis, and WHO histological type. Data on other clinical and preclinical parameters of patients and controls were also registered but are not included in this report.

Each tissue sample was stored in liquid nitrogen immediately after surgical excision for a short time and then frozen in a −80 °C refrigerator prior to processing. A piece of the tissue (20 mg ± 1.00 mg) for each sample was weighed and placed into a 2 mL centrifuge tube after thawing in a 4 °C refrigerator. To fully extract metabolites and remove the protein, a porcelain ball and 800 μL of methanol/water (8:2, *v*/*v*) solvent were added and then homogenized in a frozen mixed ball grinding machine. The homogenization took 25 s with 5 s intervals each time. Subsequently, the mixture was centrifuged at 12,000× *g* for 15 min at 4 °C. After that, 200 μL of supernatant was acquired from each sample and then frozen to dry for subsequent analysis; 20 μL of each supernatant was pooled as quality control (QC) samples to monitor the stability of the instrument. Finally, the dried samples were redissolved in 100 μL of 75% acetonitrile (ACN) for subsequent LC−MS analysis.

### 4.2. LC-MS Analysis and Data Processing

LC-MS data were obtained in positive ion mode by an ultrahigh-performance liquid chromatography (UPLC) system (1260 infinity Series, Agilent Technologies, Palo Alto, CA, USA) coupled to a quadrupole time-of-flight (Q-TOF) mass spectrometer (Agilent 6530, Agilent Technologies, Palo Alto, CA, USA). LC separation was carried out on an Agilent SB-C18 column (particle size, 1.8 µm; 100 mm (length) × 2.1 mm (i.d.)) maintained at 40 °C. Chromatographic conditions were as follows: flow rate, 0.3 mL/min; sample injection volume, 10 µL; ESI+: mobile phase A, 0.1% FA in ACN; mobile phase B, 0.1% FA in water. The linear elution gradient program was set as follows: 0–1 min: 5% B; 1–10 min: 5% B to 95% B; 10–13 min: 95% B; 13–13.1 min: 95% B to 5% B; 13.1–20 min: 5% B. The acquisition rate of MS data acquisition was set at two spectra/second, and the TOF mass range was set at m/z 50–1000 Da. Other parameters were set as follows: dry gas temperature: 350 °C; dry gas flow rate: 10 L/min; nebulizer pressure: 50 psi; Oct RFV, 750 V; fragmentor voltage: 120 V; and capillary voltage: 4000 V. A blank sample (ACN: H_2_O, 1:1, *v*/*v*) was applied after every 5 injections of the biological sample to monitor the stability of the data acquisition [38].

The raw MS data were transformed into MZML files with ProteoWizard. Extracted spectra were processed using Metaboanalyst 5.0 (http://www.metaboanalyst.ca/, accessed on 10 February 2021) for preprocessing, including peak picking, peak alignment, peak annotation, and contaminant exclusion. The peak-picking step was first performed to obtain all peaks from the entire spectrum. The centWave algorithm, which is used to detect high-resolution MS data, is supported. The subsequent peak alignment included peak grouping and retention time correction. Peak annotation, as well as the identification of molecules that can be considered potential metabolites, was achieved utilizing the Human Metabolome Database (HMDB) on Metaboanalyst 5.0. Contaminants, which are defined as peaks with a retention time range over half of the chromatogram and should be excluded from the parameter optimization step, are automatically excluded. After that, we checked the integrity of the data, filled missing values by column (feature) min value, filtered features if their relative standard deviations (RSDs) were >25% in QC samples, and finally processed the data with sample normalization (normalizing by sum) and data scaling (autoscaling) to convert the data into an approximately normal distribution. After obtaining the processed LC-MS data, we first used Simca 14.0 to perform principal component analysis (PCA) and partial least squares discriminant analysis (PLS-DA) in terms of examining the quality of the data and initially determining the metabolic differences between the groups. Subsequently, we performed a permutation test to verify that the model was not overfitted.

### 4.3. Transcriptomic Data Collection and Processing

We downloaded mRNA expression and clinical data (TCGA-THYM) from The Cancer Genome Atlas (TCGA, http://cancergenome.nih.gov/, accessed on 4 April 2022) for 117 samples of TETs. The probe identifiers in the gene matrix files were converted to gene symbols according to the annotation files of the corresponding platforms. The relevant data provided by TCGA are public and open; therefore, no additional ethical approval was required. We then extracted mRNA expression and clinical data for the genes associated with the differential metabolism obtained by LC-MS through the metabolite–protein interaction networks (MPIs) for subsequent analysis.

### 4.4. WGCNA Analysis

Weighted gene co-expression network analysis (WGCNA) was performed with the WGCNA R package using the TCGA expression files. We applied WGCNA to investigate the association between clinical traits and expression modules [39], which means WHO histological type and prognosis-related genes (TP-MRGs) in this study. Primarily, gene pairs with Pearson coefficients above a threshold were fitted into a matrix. Then, a power function was used to build the adjacency matrix. Once a weighted network was constructed, module detection was performed on clusters of closely related genes. To assess the degree of gene association in the network and to reduce spurious connections between genes, the topological overlap was inserted to identify highly similar genes. Finally, similar gene expression patterns were grouped into modules of the same color. We used the Heat Map toolkit in R to calculate correlations between gene modules and clinical features and to plot the heatmap [40].

### 4.5. Consensus Clustering for TP-MRGs

To further explore the metabolic pattern in TETs, 117 patients were clustered by ConsensusClusterPlus (resample rate of 80%, 50 iterations, and Pearson correlation, http://www.bioconductor.org/, accessed on 14 April 2022) according to 147 TP-MRGs’ expression [41]. We applied the consensus clustering algorithm to identify the number of subgroups and their stability. In addition, principal component analysis was used to explore the gene expression models of different clusters.

### 4.6. Construction of the Metabolic Score

First, LASSO regression analysis was performed on the basis of 137 TP-MRGs’ expression. After that, three genes were identified and utilized in the construction of the metabolic score. A multivariate Cox risk proportional regression model was used to determine the influence coefficient of each gene on patient survival. Finally, a formula similar to that in previous studies [42,43] was applied to define the metabolic score:$$\mathrm{Metabolic}\;\mathrm{score}=\sum GENE_{i}-\mathrm{Metabolic}\;\mathrm{score}\;\sum GENE_{j},$$
where $GENE_{i}$ is the score for genes with positive Cox coefficients, and $GENE_{j}$ is the score for genes with negative Cox coefficients.

### 4.7. Gene Set Variation Analysis and Gene Set Enrichment Analysis

Gene set variation analysis (GSVA) [44] and the “GSVA” R package were used to explore different biological pathways for different metabolic scores. We downloaded the gene set “msigdb.v7.5.1.symbols.gmt” from the MSigDB database and used it for the GSVA assay. Adjusted *p*-values of less than 0.05 were considered statistically significant. Similarly, gene set enrichment analysis (GSEA) is a computational method used to determine whether a large set of defined genomes exhibit statistically significant differences in biological status [45]. Patients were divided into two subgroups based on median expression values of prognostic metabolic signature genes (MSGs) in TETs, and KEGG gene set enrichment analysis in “msigdb.v7.5.1.symbols.gmt” was performed, with a *p*-value < 0.05 being considered statistically significant.

### 4.8. Tumor-Infiltrating Immune Cell Evaluation and ESTIMATE

We estimated the abundance of 22 different TIICs in the expression profiles of TETs using the CIBERSORT algorithm (http://cibersort.stanford.edu/, accessed on 17 April 2022) [46], which is more suitable for analyzing unknown mixed contents and noise than the previous deconvolution approach. In this study, the R package “CIBERSORT” was used to estimate the proportion of immune cells in TCGA-THYM samples. The expression data were used to estimate stromal and immune cells in malignant tumors using the ESTIMATA algorithm [47], which applies gene expression features to predict the cellular structure and purity of tumor cells in tumors. We determined stromal scores, immune scores, ESTIMATE scores, and tumor purity scores separately for each sample in TCGA-THYM using the ESTIMATE algorithm from the R “estimate package”.

### 4.9. Statistical Analysis

Statistical analyses were performed using the R language (version 4.1.2). Differences between the two groups of expression profiles were analyzed using the DESeq2 package. Survival curves were generated for each subgroup of the dataset using the Kaplan–Meier method, and log-rank (Mantel–Cox) tests were used to determine the statistical significance of the differences. We used the “survivor” R package to generate the survival curves. Univariate or multifactorial analyses were performed by relying on Cox proportional risk models using the survival package in R. Unless otherwise stated, *p* < 0.05 was considered statistically significant.

## 5. Conclusions

In this study, we screened for differential metabolic signatures between TETs and TNTs by LC-MS. Then, we accordingly developed and validated a reliable metabolic model for TETs, subsequently systematically linking these metabolic patterns to the survival characteristics of tumor patients and the characteristics of tumor immune cell infiltration. More broadly, the metabolic pathways identified in this study provide new ideas for the study of tumorigenesis in TETs; meanwhile, the two genes associated with the differential metabolic profile provide an interesting starting point for uncovering new prognostic markers or potential therapeutic targets for TETs.

## Figures and Tables

**Figure 1 metabolites-12-00567-f001:**
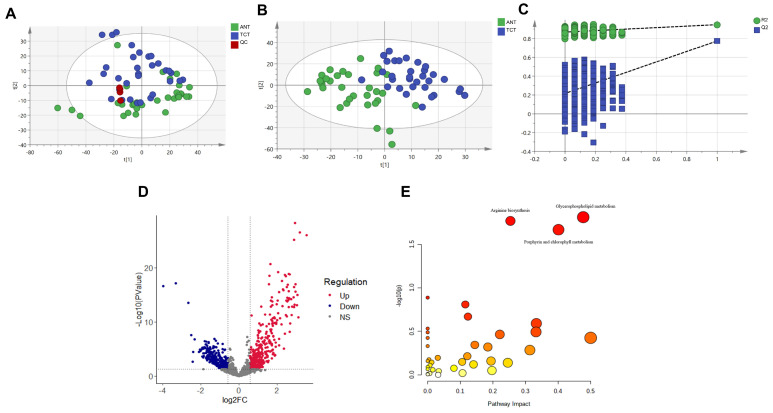
Differential metabolic features and pathways between TCTs and ANTs. (**A**,**B**) Score plots of PCA and PLS–DA for TNTs and ANTs. (**C**) Results of 1000–times permutation tests of PLS–DA model by cross–validation for TCTs vs. ANTs. (**D**) Volcano diagram revealing changes between TCTs and ANTs. The red dots on the right side of the figure represent upregulated metabolites, the blue dots on the left side mean downregulated metabolites, and the black dots refer to non-significant differences. The *x*-axis corresponds to the log2-fold change, and the *y*-axis corresponds to the –log10 *p*-value. (**E**) Enrichment pathways of differential metabolic features between TCTs and ANTs. The color of the circle indicates the significance level in the enrichment analysis, where a darker color indicates greater significance (pathways with *p* < 0.05 were annotated); the size of the circle reflects the pathway impact value in the topology analysis, where the larger the circle, the larger the impact value. The *x*-axis is the pathway impact value calculated based on topology analysis. TNT: thymic cancerous tissue; ANT: adjunct noncancerous tissue; QC: quality control; NS: non-significant.

**Figure 2 metabolites-12-00567-f002:**
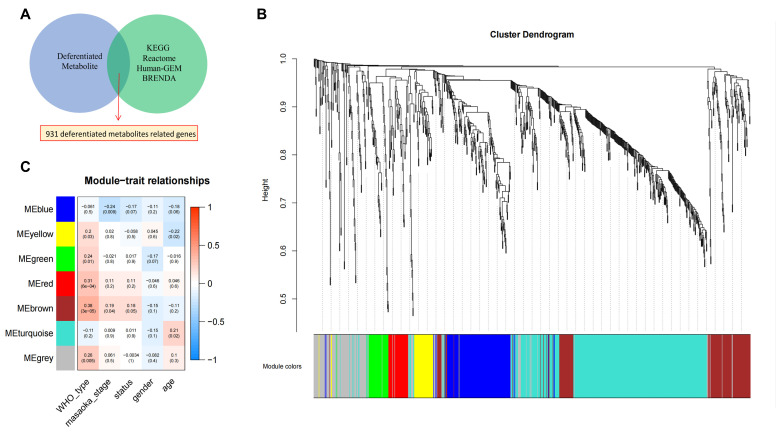
Identification of metabolite-related genes and WGCNA. (**A**) Venn diagrams of differentiated metabolite-related genes from metabolite–protein interactions. (**B**) Hierarchical clustering tree for 931 MRGs. (**C**) Module–trait relationship heatmap based on the Pearson correlation coefficient between module eigengenes and clinical parameters (WHO histological type, Masaoka stage, survival status, sex, and age).

**Figure 3 metabolites-12-00567-f003:**
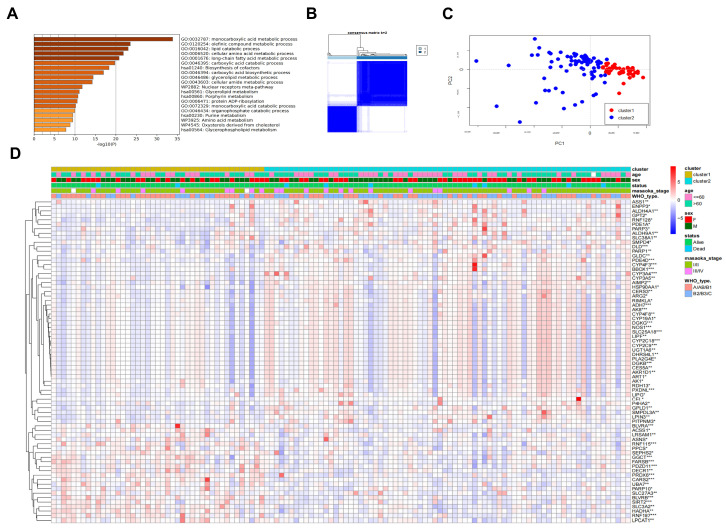
Differential clinicopathological features and enriched pathways of thymic epithelial tumors in the cluster ½ subgroups. (**A**) Gene ontology biological process and pathway enrichment analysis of the TP-MRGs. (**B**) Consensus clustering matrix for k = 2. (**C**) Principal component analysis of gene expression patterns between the two subtypes. (**D**) Heatmap and clinicopathologic features of the two clusters (clusters ½) defined by the MRG consensus expression. * *p* < 0.05, ** *p* < 0.01, *** *p* < 0.001.

**Figure 4 metabolites-12-00567-f004:**
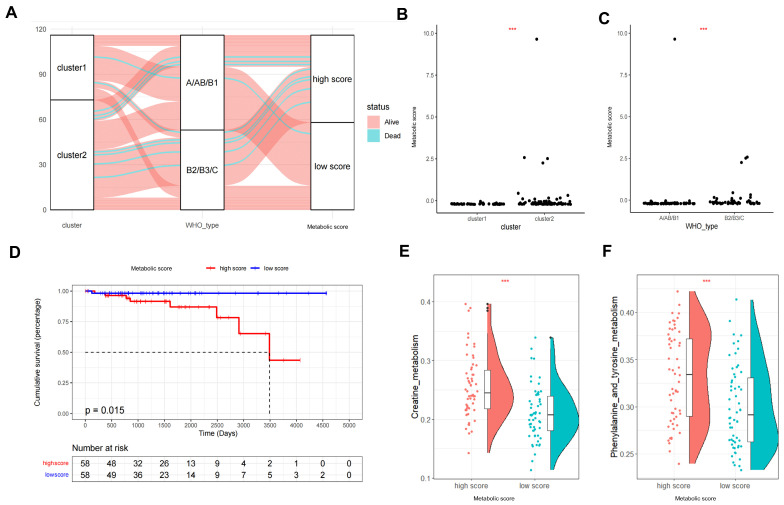
Construction of the metabolic score and exploration of the correlations of clinical features and biological pathways. (**A**) Alluvial diagram of groups in different TP-MRG clusters, WHO histological types, metabolic scores and survival status. (**B**,**C**) Differentiated metabolic scores between different TP-MRG clusters and WHO histological types (Wilcoxon test). (**D**) Kaplan–Meier curves for high- and low-metabolic-score groups of thymic epithelial tumors. (**E**,**F**) Raincloud plots showing the two differential metabolic pathways in high- and low-metabolic-score groups. *** *p* < 0.001.

**Figure 5 metabolites-12-00567-f005:**
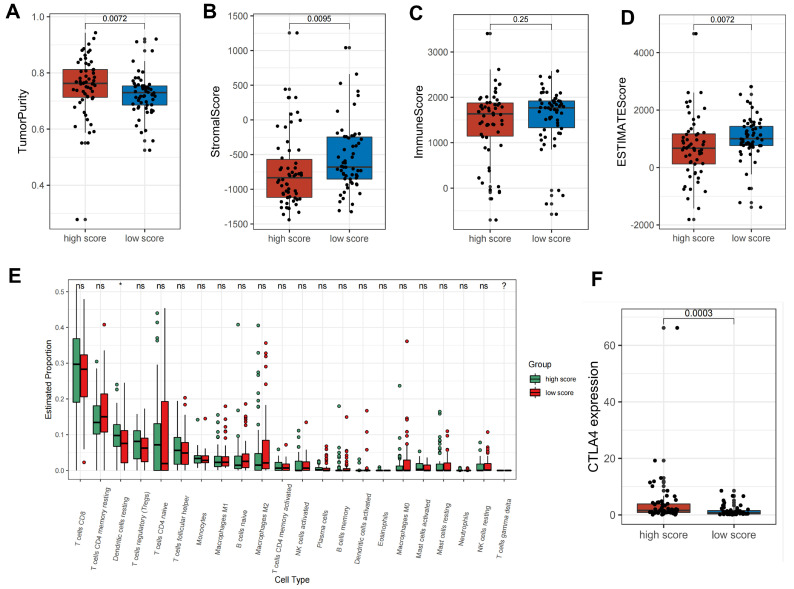
Distinct immune landscapes of metabolic patterns. (**A**–**D**) Box plots of tumor purity scores, stromal scores, immune scores and estimate scores in high- vs. low-metabolic-score groups. (**E**) The infiltrating levels of 22 immune cell types in high- vs. low-metabolic-score groups. (**F**) Comparison of the CTLA4 expression levels between the two metabolic score groups. ns: non-significant; * *p* < 0.05.

**Figure 6 metabolites-12-00567-f006:**
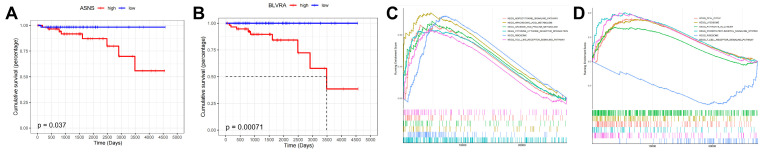
Prognostic value and GSEA of ASNS and BLVRA in thymic epithelial tumors. (**A**,**B**) Kaplan–Meier curves for groups with high and low ASNS/BLVRA expression in thymic epithelial tumors. (**C**,**D**) Gene set enrichment analysis (GSEA) showing the differential biology pathways in high and low ASNS/BLVRA expression groups.

**Table 1 metabolites-12-00567-t001:** Univariate and multivariate analyses of the correlation of clinical variables and expression of metabolic genes with overall survival in TETs.

Clinical Factors	Variables	Univariate Analysis		Multivariate Analysis	
		*p*-Value	HR (95% CI)	*p*-Value	HR (95% CI)
WHO type	A/AB/B1 vs. B2/B3/C	0.83	0.86 (0.21–3.5)	0.65	0.66 (0.11–4)
Masaoka stage	I/II vs. III/IV	0.54	1.6 (0.34–8)	0.31	2.7 (0.4–18)
Sex	Male vs. Female	0.53	0.66 (0.18–2.5)	0.62	1.5 (0.31–7.2)
Age	≤60 vs. >60 (years old)	0.17	3 (0.63–15)	0.14	4 (0.63–25)
ASNS	Expression (high vs. low)	0.071	0.15 (0.018–1.2)	0.23	0.25 (0.027–2.4)
BLVRA	Expression (high vs. low)	1	1.3 × 10^−9^ (0–Inf)	1	5.2 × 10^−10^ (0–Inf)

**Table 2 metabolites-12-00567-t002:** Patient characteristics.

Age, Years	
Median (range)	53.5 (32–80)
Sex (%)	
Female	14 (43.8)
Male	18 (56.2)
WHO histological type (%)	
A	2 (6.3)
AB	12 (37.5)
B1	1 (3.1)
B2	4 (12.5)
B3	3 (9.3)
C	8 (25.0)
NA	2 (6.3)
Myasthenia gravis (%)	
(+)	10 (31.3)
(−)	20 (62.5)
NA	2 (6.2)

Abbreviations: WHO, World Health Organization; NA, not applicable.

## Data Availability

The data presented in this study are available in online repositories (TCGA: https://portal.gdc.cancer.gov/, accessed on 4 April 2022) or openly available in the article.

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
