# Peer review of "Metabolomic and Transcriptomic Profiling Identified Significant Genes in Thymic Epithelial Tumor"

_metabolites, 2022, doi:10.3390/metabo12060567_

Round 1

Reviewer 1 Report

1. More details are required considering the data pre-processing (from mzml to metaboanalyst file).

2. In Figure 1E, only three pathways were annotated. Were there the only statistically significant ones? If not, please manually annotate all significant pathways. Besides, the detailed parameters of the pathway analysis using metabolic features should be mentioned as it predominantly affects the findings.

3. In Figure 2A, 931 differentiated metabolites-related genes were defined. The details should be given for better evaluation. In line 466, it is written that gene symbols were used. However, using gene symbols for the Venn analyses could lead to an inappropriate list of the final list of genes. It is better to convert gene names from different platforms to a unified Entrez gene ID for such a task.

4. The analyses that led to the conclusions in Figure 4 should be clarified better.

4a. The statistical significance in Figures 4B and 4C could be due to the outlier. Please explain the statistical test used.

4b. It is written that "As illustrated in the heatmap, higher metabolic scores 218 were significantly associated with activation of metabolic pathways..." (Figure 4E). Unfortunately, I cannot see a difference when looking at the heatmap. Perhaps a "group-level visualization" could do the job better.

5. Since this study links different omics platforms and has several distinct sections, it is recommended to have a figure serving as the study workflow to facilitate the readers' understanding.

6. It is better to deposit the raw or processed metabolomic data to allow reuse and reproducible analyses by other groups.

7. The limitations of the study should be mentioned. For example, lack of metabolite annotation using MS/MS and/or authentic standards.

8. In the informed consent statement, it is written that the patient consent was waived due to "medical records/biological samples in this study were obtained in previous clinical procedures." It seems not appropriate to say so. The patient has the right to be fully aware of how their information/biological samples would be used and must approve it in informed consent.

Author Response

We highly appreciate your constructive comments and suggestions on our manuscript, which are really helpful for us. We have studied the comments very carefully and have accordingly made revisions marked using “Track Changes” function in the revised version (MS Word), which we would like to submit for your consideration. A version (pdf) with the revision track hidden has also been uploaded. Here are our point-by-point responses to your comments and we hope that the revision and correction will meet with approval. The number of lines attached to each amendment is according to the version with changes track (MS Word).

1, We have added a detailed description of the data pre-processing (lines 471-479).

2, The three annotated pathway analysis in figure 1E are the only statistically significant ones (p<0.05). Therefore, we have added the explanation (pathways with P<0.05 were annotated) in the figure annotation (line 93). Also, the detailed parameters for each differential pathway have been uploaded in supplementary table S1.

3, Thank you very much for your comment. In fact, we have listed the 931 defined differential metabolism-related genes in supplementary table S2 if that is the detail needs mentioned to be provided. We do agree with the problems that may arise from using gene symbols to convert gene names from different platforms, and Entrez gene ID is more appropriate for this task. However, the article cited in this study on creating metabolite protein interaction networks for matching out metabolism-related genes only provides gene symbols, so the Entrez gene ID was unfortunately not utilized in this study.

4, Thank you very much for your suggestion. 4a: We have followed your suggestion and the statistical tests used in Figures 4B and 4C have been described in the text (line 214).  4b: We have utilized the raincloud plots to show the two metabolic pathways with significant differences, and have also amended the relevant explanation in the text and figure annotation and placed the heatmap in the supplementary figure S3.

5, Thanks for your suggestion, we have produced and uploaded the graphical abstract (line 28).

6, The processed metabolomics data has been submitted as supplementary table S4.

7, Thanks, we have added a description of the limitations of the study at the end of the discussion (lines 407-417)

8, Thank you for your comments and we totally agree with you. However, as several patients could not be contacted to obtain informed consent, we applied to the ethics committee for a waiver of informed consent. In addition, as the specimens used in this study are all tissue specimens left over from the normal course of treatment, and the study does not involve additional risk to patients, personal privacy or commercial interests, the Ethics Committee has granted a waiver of informed consent in accordance with the relevant ethical regulations and guidelines for the reasons already mentioned above and others. Notwithstanding the above, we have removed the clarification that may cause misunderstanding.

Reviewer 2 Report

The authors present their findings on the metabolomic of thymic epithelial tumors and conclude that there is a differential expression of metabolites that is associated with tumorigenesis and prognosis.

The work is innovative and uses a different technology that has not been explored in TET.

There are however a few basic questions that raise concerns about the interpretation of their findings.

1- Thymic epithelial tumor is a heterogenous group of tumor the include thymoma, thymic carcinoma and neuroendocrine tumor. There is no mention of the later. It is best to refer specifically to thymomas and carcinoma instead of TET.

2 Thymomas and thymic carcinomas are very different tumors that should not be grouped together. if the authors separate thymomas and thymic carcinomas in their analysis, would they find the same results? Although the number of cases evaluated is small, can the authors provide analysis for these two different tumors separately?

3-The WHO classification of tumor, no longer uses the terminology type C for thymic carcinomas, because they are very different.

 What type of thymic carcinoma was evaluated? thymic squamous cell carcinomas? any other type?

has the relevant metabolites identified in TET associated with squamous cell carcinoma from other sites? considering that squamous cell carcinoma is the most common type of thymic carcinoma.

4- table 1, it is interesting that the authors did not see any association of tumor type or stage with survival. This raises the question, how many patients died? Did they die of the disease or was it overall survival?

5- how were the samples collected for analysis? how long was the ischemic time from collection to freezing the specimens? an issue that is very relevant to the type of analysis performed. 

Author Response

We highly appreciate your constructive comments and suggestions on our manuscript, which are really helpful for us. We have studied the comments very carefully and have accordingly made revisions marked using “Track Changes” function in the revised version (MS Word), which we would like to submit for your consideration. A version (pdf) with the revision track hidden has also been uploaded. Here are our point-by-point responses to your comments and we hope that the revision and correction will meet with approval. The number of lines attached to each amendment is according to the version with changes track (MS Word).

1, Thank you for your comments, we have made the revision in the abstract and text respectively (lines 9-10, 33-35).

2, We thank you very much for your comments and have tried to screen separately for differential metabolism between thymoma or thymic carcinoma and their adjacent normal tissues, but unfortunately we did not see much difference between the results of thymoma and thymic carcinoma when analyzed separately and together (lines 130-135, Figure S1).

3, We have modified the description of "type C" mentioned in the text (lines 34,73-74). Four of the eight patients with thymic carcinoma included in this article were squamous cell carcinomas; however, because of the very limited number of cases, further exploration of the metabolic features specific to squamous cell carcinoma in these cases yielded very limited results and therefore is not shown in the text.

4, As for the association between the variables and survival, the clinical data we applied derived from the TCGA consisting of 117 samples, of whom nine died during follow-up. The survival data provided by this data are overall survival. Thus, it’s believed that the lack of association may still be due to the limited number of cases in the sample.

5, Thanks for your advice. We have added methods on sample collection and storage (lines 438-439).

Reviewer 3 Report

The article titled  ‘Metabolomic and Transcriptomic Profiling Identified Significant Genes in Thymic Epithelial Tumormay’ may be an useful contribution to the journal; the research is sound, adds some elements of novelty; the manuscript is thoroughly documented and the methodology is adequately chosen; the perspective is fresh and updated. The conclusions are supported by the results.

A Limitations section to provide the limitations of the article could help the reader and could be included in the end of the Discussion section (encompassing various limitations such as: e.g. the reduced number of cases included in the study, unicentricity of the study, among others).

Grammar and punctuation must be carefully checked within the entire article 

Author Response

We highly appreciate your constructive comments and suggestions on our manuscript, which are really helpful for us. We have studied the comments very carefully and have accordingly made revisions marked using “Track Changes” function in the revised version (MS Word), which we would like to submit for your consideration. A version (pdf) with the revision track hidden has also been uploaded. Here are our point-by-point responses to your comments and we hope that the revision and correction will meet with approval. The number of lines attached to each amendment is according to the version with changes track (MS Word).

Thank you very much for your suggestions. We have added the limitations of the article at the end of the discussion, and we have also checked the grammar and punctuation of the entire text with care and made separate changes.

Round 2

Reviewer 1 Report

The quality of the manuscript has been improved significantly thanks to the clarifications made by the authors.

I still can't entirely agree with the argument about using gene names and bypassing the Entrez. This technical decision might influence the findings. However, it should not significantly affect the overall conclusions.

Regarding the concern related to the IRB, the authors followed the local regulations. Thus, I would leave the responsibility to them if any issues that might arise in the future.